# Hope For The Best But Prepare For The Worst: Cautious Adaptation In RL Agents

## Abstract

We study the problem of safe adaptation: given a model trained on a variety of past experiences for some task, can this model learn to perform that task in a new situation while avoiding catastrophic failure? This problem setting occurs frequently in real-world reinforcement learning scenarios such as a vehicle adapting to drive in a new city, or a robotic drone adapting a policy trained only in simulation. While learning without catastrophic failures is exceptionally difficult, prior experience can allow us to learn models that make this much easier. These models might not directly transfer to new settings, but can enable cautious adaptation that is substantially safer than naïve adaptation as well as learning from scratch. Building on this intuition, we propose risk-averse domain adaptation (RADA). RADA works in two steps: it first trains probabilistic model-based RL agents in a population of source domains to gain experience and capture epistemic uncertainty about the environment dynamics. Then, when dropped into a new environment, it employs a pessimistic exploration policy, selecting actions that have the best worst-case performance as forecasted by the probabilistic model. We show that this simple maximin policy accelerates domain adaptation in a safety-critical driving environment with varying vehicle sizes. We compare our approach against other approaches for adapting to new environments, including meta-reinforcement learning.

## 1 Introduction

An experienced human driving a rental car for the first time is initially very aware of her lack of familiarity with the car. How sensitive is it to acceleration and braking? How does it respond to steering? How wide is the vehicle and what is its turning radius? She drives mindfully, at low speeds, braking far ahead of desired stops, and making wide turns, all the while observing the car's responses and adapting to it. Within minutes, once she is familiar with the car, she begins to drive more fluently and efficiently. Humans draw upon their prior experiences to perform this kind of safe, quick adaptation to unfamiliar situations all the time, such as when playing with a new tennis racquet, or walking on a new slippery surface.

Such problems are critical to address in autonomous systems: such as when a self-driving car must learn to drive in a new country, or when a planetary rover might have to learn to explore a harsh new environment. Missteps in real-world situations can cause real damage to robots and their environments. An important bottleneck in applying today's standard machine learning approaches to control in these real-world situations is that they are trained without any notion of safe behavior under uncertainty. Recent works have attempted to address this by proposing methods for safe exploration during reinforcement learning — in other words, how might an agent avoid risky actions during training time? This still requires that the robot acquire its notions of uncertainty and risks at the same time as it is learning to perform tasks in the new environment, which is difficult and precarious.

Could we instead rely on transferring notions of uncertainty and risk acquired from prior experience in other related domains, such as in simulated environments, where safety may not be as much of a concern? In other words, could we make the safe learning problem easier through knowledge transfer, relaxing the problem to *safe adaptation*, like the human driver? How might the planetary rover draw on its experience in many varied terrains on Earth to perform meaningfully cautious actions during learning on the unknown terrain of a new planet?

Motivated by these questions, we propose a model-based reinforcement learning approach called risk averse domain adaptation (RADA). RADA works by first pretraining a probabilistic dynamics model on a population of training domains with varied, unknown dynamics. Through this experience over many environments, the model learns to estimate the epistemic uncertainty (model uncertainty) of unknown environment dynamics, thus permitting estimation of a distribution of outcomes for any action executed by the agent. When introduced into a new target environment, RADA uses this estimated distribution of outcomes to select *cautious* actions that obey the following maximin notion of risk-aversion: among various candidate action sequences, it executes those that lead to the best *worst-case* performance, as predicted by the model. Much like the human driver in the example above, all the information collected during this cautious phase of exploration is fed back into the model to finetune it to the new domain, leading to increasingly confident predictions. Over time, RADA steadily estimates lower risks and approaches optimality in the target environment. As we demonstrate in experiments in a driving domain, the experience acquired during RADA's pretraining phase enables fast yet safe adaptation within only a handful of episodes.

## 2    RELATED WORK

Cautious or risk-averse learning has close connections to learning robust control policies, as well as the uncertainty estimation derived from Bayesian reinforcement learning (Ghavamzadeh et al., 2015; Strens, 2000). Rather than conventionally maximizing a reward function, accounting for risk usually involves allocating more attention to 'worst-case' outcomes in an environment. Such outcomes become particularly important in out-of-domain settings, where purely optimizing in the training domain does not guarantee good performance in the test domain, the problem setting that we consider in this work.

**Safety in reinforcement learning.**    Incorporating safety requires properly managing risks and reducing the impact of unforeseen negative outcomes. Risk management is extensively studied in quantitative finance. In portfolio optimization, a commonly used quantity that measures the expected return considering the worse $\alpha$-% of cases is Conditional Value at Risk (CVaR) (Rockafellar et al., 2000). With probability $\alpha$, the reward is greater than the CVaR measure. CVaR is formulated as $E[R|R \leq \upsilon_\alpha]$. Rather than optimizing the expected reward, risk averse policies optimize the lower $\alpha$-quartile of the distribution of rewards.

While meta-learning approaches like $RL^2$ (Duan et al., 2016) can potentially learn safety by adapting across learning episodes, we found this was not possible in the environments we tested. To address safety more expicitly, the reinforcement learning community is adopting measures like CVaR as quantities that can be optimized (Morimura et al., 2010; Borkar & Jain, 2010; Chow & Ghavamzadeh, 2014; Tamar et al., 2015; Chow et al., 2015) to create policies which are robust to shifts from source to target domains. Rajeswaran et al. (2016) propose learning robust policies by sampling from the $\alpha$-quartile of an ensemble of models. While the model ensemble is trained on a given source distribution, the policy is only trained on the lower $\alpha$-quartile rewards from trajectories sampled from this ensemble. This leads to policies which are more conservative and therefore more robust to shift when deployed from source to target domains.

**Epistemic uncertainty in reinforcement learning.**    While learning a robust model is beneficial for transferring to different domains, model-based reinforcement learning offers an additional unsupervised learning signal that can be exploited at test time. In particular, prior work has shown that a model can be quickly adapted during test time by meta-learning for fast adapting parameters during training (Nagabandi et al., 2018a; Sæmundsson et al., 2018). These fast adapting parameters offers greater flexibility in adapting to unforeseen circumstances which an agent may encounter at test time. Nagabandi et al. (2018a) show that real robots can quickly adapt to broken or miscalibrations during evaluation through this fast adaptation acquired through meta-learning. Such approaches are complementary to our approach, as they provide a means to explicitly train for fast adaptation to disturbances in the environment, while they do not account for any notion of safety.

Henaff et al. (2019) propose using the uncertainty of a model to regularize policy learning. The policy is encouraged to create trajectories which are distributionally close to trajectories observed in training data. After training, the observations of the model and actions of the policy generate state trajectories which near the domain it has been trained on. In other words, the policy has a preference to keep the trajectories within its training domain. In our work, our policy is encouraged to behave

cautiously in unfamiliar environments rather than remain in familiar ones. Kahn et al. (2017) train a collision prediction model to favor 'safe' (low velocity) collisions. Using uncertainty estimates, this collision model will initially behave cautiously in a new environment. Similar to our method, the model becomes less conservative as it adapts to the new environment and lowers its uncertainty.

**Domain randomization.**   Domain randomization (Sadeghi & Levine, 2017; Peng et al., 2018; Tobin et al., 2017) attempts to train policies that are able to transfer to some target environment by training an agent in deliberately randomized simulated environments to allow learning a policy that is *invariant* to the randomized parameters, and thus performs robustly in the target environment. RADA also pretrains on a set of environments with varied dynamics, but different from these prior works, we operate in a safety-critical setting, focusing on safe adaptation to the target environment — to accomplish this, we follow an explicitly cautious action policy at adaptation time, different from the policy used in the pretraining environments.

Before discussing RADA, we first lay out some preliminaries.

## 3   BACKGROUND: PETS

We build upon PETS (Chua et al., 2018), a recently proposed approach for model-based reinforcement learning. We describe the main features of the PETS framework below:

**Probabilistic dynamics model.**   PETS trains an ensemble of probabilistic dynamics models within its environment. Each model in the ensemble is a probabilistic neural network that outputs a distribution over the next state $s'$ conditioned on the current state $s$ and action $a$. The data for training these models comes from trajectories executed by following the same scheme for action selection that will be eventually used at test time.

**Action selection.**   This action selection scheme is sampling-based model-predictive control (MPC): an evolutionary search method finds action sequences with the highest predicted reward. The reward of an action sequence in turn is computed by propagating action outcomes autoregressively through the learned probabilistic models.

**Reward computation.**   Specifically, starting from a state $s_0$, for each sampled action sequence $A = [a_1, ..., a_H]$, where $H$ is the planning horizon, the dynamics model first predicts a distribution over $s_1$ after executing $a_0$. A particle propagation method samples Monte Carlo samples from this distribution. For each sample, the dynamics model then predicts the state distribution for $s_2$, conditioned on executing $a_1$, and the process repeats. This recursive particle propagation results in a large number $N$ of particles $\{\hat{s}_H^i\}_{i=1}^N$ after $H$ steps. These $N$ particles represent samples from the distribution of possible states after executing $A$. Each such particle $i \in [1, N]$ is now assigned a predicted reward $r^i$, which is a function of its full state trajectory starting from $s_0$. Finally, the mean of those predicted rewards is considered the score of the action sequence:

$$R(A) = \sum_i r^i / N. \tag{1}$$

We call this the action score. Then, the winning action sequence $A^* = \arg\max_A R(A)$ with the highest action score is selected, the first action in that sequence is executed, and the whole process repeats starting from the resulting new state $s_1$.

## 4   RADA: RISK-AVERSE DOMAIN ADAPTATION

Now we present our approach, Risk-Averse Domain Adaptation (RADA). As motivated in Sec 1, RADA approaches safe learning as an *adaptation* problem, where an agent may draw upon its experience in a variety of environments to guide cautious behavior in a new safety-critical target environment while minimizing the risk of catastrophic failure.

Consider a set of environments, each defined by the value of an unknown domain ID variable $z$, which controls the dynamics in each domain. RADA assumes a setting where we train on some of these domains, and must then transfer to new domains with unknown, potentially unseen, and even out-of-distribution values of $z$. As a running example, consider learning to drive cars, where each car is represented by its own value of the domain ID variable $z$. This domain ID might include a

potentially large set of unknown and hard-to-measure properties, which together determine the way that the car drives.

We propose a solution, RADA, that builds upon the PETS framework (Sec 3). PETS has been demonstrated to work well across a variety of environments. Further, compared to alternative approaches, this framework has two natural advantages for our risk-averse domain adaptation setting, both of which are critical to our approach: (i) the probabilistic models of PETS can be adapted to capture the "epistemic uncertainty" about the dynamics of a new domain, and (ii) model-based RL agents contain dynamics models that can be trained in the absence of any rewards or supervision, providing a route for adaptation to a new environment. We now discuss how RADA builds upon PETS.

RADA first builds a probabilistic ensemble of dynamics models on the training domains that capture the epistemic uncertainty in dynamics due to unknown $z$. We call this the pretraining phase. When an agent with this pretrained model encounters a new domain, we use pessimistic predictions from the model to drive cautious exploration to finetune the model to the new domain, leading to safe and fast adaptation. We call this the adaptation/finetuning phase. Algorithm 1 provides pseudocode for RADA, and the rest of this section explains RADA in detail.

## 4.1 PRETRAINING PETS IN MULTIPLE DOMAINS

While the PETS probabilistic ensemble is trained to represent uncertainty within a single environment in Chua et al. (2018), we would like our model to capture the uncertainty associated with being dropped into a new environment, with unknown domain ID $z$.

To do this, we propose a "pretraining" phase, where a single PETS ensemble is trained across all the training environments, with varying, unknown values of $z$. Specifically, at the beginning of each training episode, we randomly sample one of the training $z$'s from a uniform distribution. Since $z$ determines the environment dynamics and is unavailable to the learned dynamics model, the ensemble has incentive to learn to model this as epistemic uncertainty during the pretraining phase. See the pretraining procedure in Algorithm 1.

## 4.2 CAUTIOUS ACTION SELECTION DURING ADAPTATION

After this pretraining, how might the uncertainty captured in the ensemble inform cautious exploration during adaptation in the target environment? To do this, we adapt the PETS action selection and reward computation scheme using a maximin notion of cautious behavior, in line with notions of risk used in prior work across disciplines (Rockafellar et al., 2000; Tamar et al., 2015; Rajeswaran et al., 2016).

Specifically, we replace the action score of equation 1 with a newly defined "generalized action score" $R_\gamma(A)$, in which the "caution parameter" $\gamma \in [0, 100]$ controls the degree of caution exercised in evaluating action sequences in the new environment. $R_\gamma(A)$ is defined as:

$$R_\gamma(A) = \sum_{i:r^i \leq \upsilon_{100-\gamma}(r)} r^i/N, \tag{2}$$

where $\upsilon_k(r)$ denotes the value of the $k^{th}$ percentile of predicted rewards $\{r^j\}_{j=1}^N$ among the $N$ particles after particle propagation.

Unpacking this definition, $R_\gamma$ measures the mean score of the bottom $100 - \gamma$ percentile of the predicted outcomes from the PETS model. When $\gamma = 50$, for instance, it measures the mean of the worst 50 percentile of predicted rewards. This is a pessimistic evaluation of the prospects of the action sequence $A$ — it only pays attention to the worst performing particles in the distribution. At caution $\gamma = 0$, $R_\gamma$ exactly matches the definition of $R$ in equation 1: it measures the mean predicted reward of all the particles produced after particle propagation through the model. In our experiments, we heuristically set $\gamma = 50$.

Now, we define a "$\gamma$-cautious action policy" as one that selects actions based on the generalized action score $R_\gamma$. In other words, $A_\gamma^* = \arg\max_A R_\gamma(A)$. We propose to deploy such $\gamma$-cautious action policies at adaptation time.

The intuition is straightforward: in an unknown environment, the agent deliberately performs cautious actions to perform safe adaptation. Even though it eventually seeks to achieve high mean performance in the target environment, it does not select actions that achieve the highest expected reward under its model. Instead, it chooses to be conservative, not trusting the model's expectations fully.

### 4.3 MODEL FINETUNING

As it gathers experience in the target environment using the $\gamma$-cautious policy, RADA also improves its dynamics model over time to finetune it to the new environment. Since dynamics models do not need any manually specified reward function during training, the ensemble model can continue to be trained in the same way as during the pretraining phase.

We propose to stabilize adaptation by keeping the model close to the original model. To do this, we maintain a replay buffer of data from the pretraining episodes, conducted outside the target domain. During adaptation, we compute all model updates on this full dataset, including the replay buffers. We use probabilistic neural network ensembles for the dynamics model Chua et al. (2018), and training proceeds through stochastic gradient descent.

As the model improves over time, the distribution of predicted outcomes becomes more and more narrow over time. For a deterministic environment, the model eventually converges to deterministic predictions, so that $R_\gamma$ is the same for all $\gamma$. In other words, once the model is well-trained, the $\gamma$-cautious action policy is identical to the standard action policy. The adaptation procedure in Algorithm 1 sums up cautious action selection and model finetuning.

---

**Algorithm 1** RADA

1: **procedure** PRETRAINING
2:     Initialize the probabilistic ensemble dynamics model $f$
3:     Initialize data $\mathcal{D}$ using a random controller in a random training domain for one trial.
4:     **for** domain ID $z \sim$ training domains **do**
5:         Train the probabilistic ensemble dynamics model $f$ on $\mathcal{D}$
6:         **for** $t = 0$ to task horizon **do**
7:             **for** evolutionary search stage=1,2,... **do**
8:                 **for** sampled action sequence $A$ **do**
9:                     Run state propagation to produce $N$ particles
10:                     Evaluate $A$ as $R(A) = \sum_i r_i / N$
11:                 **end for**
12:                 Refine search to find $A^* = \arg\max R(A)$
13:             **end for**
14:             Execute first action of $A^*$
15:             Record outcome in $\mathcal{D}$
16:         **end for**
17:     **end for**
18: **end procedure**
19: **procedure** ADAPTATION($\mathcal{D}$,f)
20:     **for** target domain adaptation episode=1,2,... **do**
21:         **for** $t = 0$ to task horizon **do**
22:             **for** evolutionary search stage=1,2,... **do**
23:                 **for** sampled action sequence $A$ **do**
24:                     Run state propagation
25:                     Evaluate $A$ with generalized score $R_\gamma(A)$
26:                 **end for**
27:                 Refine search to find $A^* = \arg\max R_\gamma(A)$
28:             **end for**
29:             Execute first action of $A^*$
30:             Record outcome in $\mathcal{D}$
31:             Finetune the probabilistic ensemble model $f$ on $\mathcal{D}$
32:         **end for**
33:     **end for**
34: **end procedure**

---

## 5 EXPERIMENTS

We now evaluate various ablations of RADA in a driving environment, evaluating the importance of three components of our technique for generalization to unseen out-of-domain environments: (i) pretraining on multiple domains (i.e. 'domain randomization'), (ii) replaying pretraining to stabilize finetuning in the target environment, (iii) $\gamma$-cautious action policy ($\gamma > 0$) at adaptation time — we heuristically set $\gamma = 50$ for our experiments. RADA is the version that includes all three techniques. For all methods, we use a PETS ensemble of 5 fully connected models, with each one having 4 layers. We hypothesize that being cautious would not only allow RADA to adapt quicker by avoiding risk of catastrophic failure, but also that as uncertainty about the environment is resolved during adaptation, a cautious policy will become similar to the optimal policy, leading to higher final reward.

**Baselines.** Our first baseline is RADA without cautious adaptation in the test environment: `RADA\caution`. Next we separately ablate out multi-domain pretraining and pretraining replay from this approach to create two more baselines: `RADA\{caution,DR}` and `RADA\{caution,replay}`. Through these baselines, we systematically compare the contributions of various components of RADA. We also compare RADA against PETS trained directly in the target environment, `train-on-target`.

As an external baseline, we compare against robust adversarial reinforcement learning (`RARL`) Pinto et al. (2017). `RARL` works by training a model-free RL agent jointly with an adversary that perturbs the actions emitted by the agent. We train the adversary to perturb the motor torques in Duckietown.

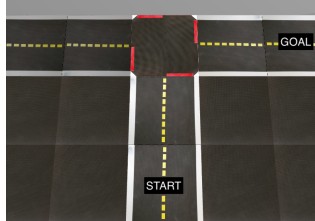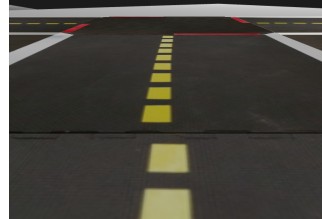

Figure 1: (left) Overhead and (right) first-person view from the car camera, of the Duckietown-based driving environment.

Finally, we also implemented three meta-learning approaches as baselines: GrBal (Nagabandi et al., 2018a), $RL^2$ (Duan et al., 2016), and MOLe (Nagabandi et al., 2018b). However, in our experiments, all three metalearning approaches experienced training failures in our environment, failing to consistently reach the goal even during the pretraining phase. Consequently, we do not consider these methods in the more detailed results reported in this section.

**Car driving environment.** Our driving environment is based on Duckietown (Chevalier-Boisvert et al., 2018), a physically accurate driving environment designed for sim-to-real transfer. The task, illustrated in Fig 1, is to make a right turn around a corner to reach a fixed goal tile. Each tile is fixed to a size of 0.585. We modify the environment such that when the car attempts to take an action that would drive it off of the road tiles, it is held in place. If the car clips the corner, it gets stuck unless the agent has learned to reverse out from the corner and try the turn again. The task rewards faster, more efficient turns, so that there is incentive to go near the corner. At the same time, since there is a big price to pay for it, a cautious agent must avoid hitting the corner at all costs. The agent can observe its current $x, y$ coordinates, current velocity, and steering angle. At each time step, it is given a reward equal to negative Manhattan distance from the goal, with a completion bonus of 100 if it successfully reaches the goal tile. The agent has direct control of the steering angle and velocity of the car at each time step. See figure 1.

Each domain is specified by a one-dimensional domain ID, the width of the car, which is unknown to the agent. During pretraining, the car width ranges from 0.050 to 0.099, and is sampled uniformly from this range before each training episode. Having driven these cars of varying widths, we evaluate each method's ability to adapt to driving new cars. We test adaptation to one in-distribution domain: width 0.075, and five out-of-distribution domains: 0.1, 0.125, 0.15, 0.175, and 0.20. We vary the car width because of its direct influence on the optimal trajectory of the car: wider cars must make wider turns to avoid catastrophe.

**Performance Metrics.** We measure the return (sum of rewards over a full episode) and the number of collisions underwent in the target environment. For each method, we report the "average maximum reward" over adaptation time $t$, which is the average over 10 random seeds of the maximum over $t$ adaptation episodes of the reward obtained in the target environment. Finally, to measure the safety of the adaptation process, we also report the cumulative number of collisions suffered by each method during adaptation, which more directly measures the extent to which different methods avoid catastrophic failures and perform safe adaptation.

**Results.** For all RADA variants, we perform pretraining for 32 episodes: 2 initial rollouts with a random policy and 30 on-policy episodes. RADA\{caution,DR} is pretrained for the same 32 episodes but on a single training domain — the one with car width 0.1, which is the closest to the out-of-domain target car widths (recall the training car widths are 0.050-0.099). RARL trains model-free policies, which typically require more pretraining episodes: we use the default settings and train it for 1000 episodes in the pretraining environments.

Fig 2 shows the average maximum reward after 10 adaptation episodes and the average total number of collisions for each method, as a function of the car width. All methods perform worse farther away from the training car widths, but RADA maintains its performance up to car width 0.15, and deteriorates more gracefully, maintaining good performance even up to car width 0.2, over two times the largest car width observed at pretraining time. Not only does RADA achieve the highest rewards,

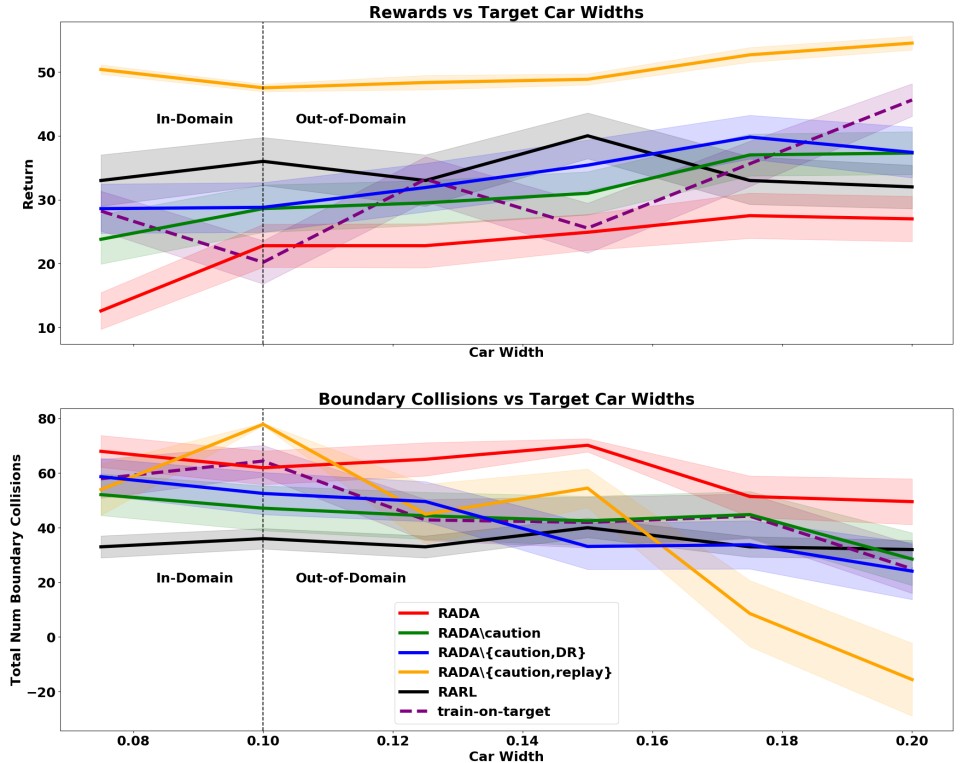

Figure 2: Evaluation of the average maximum reward and average total boundary collisions at different target car widths.

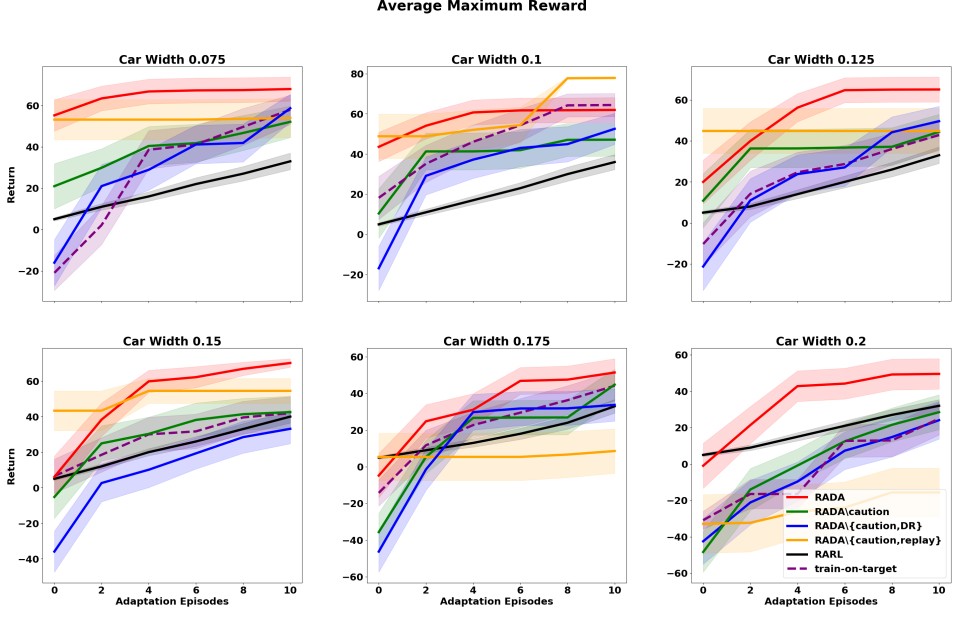

Figure 3: Evaluation of the reward over time at six different car width domains. For every evaluation, we perform ten adaptation steps, with one initial evaluation before adaptation starts. Each calculated reward is calculated by taking the maximum reward seen so far at each timestep, where the value at that timestep is averaged over ten evaluations. This maximum is then averaged over ten initializations for each model. Standard errors are shown with colored bars.

**Average Total Number of Boundary Collisions**

Figure 4: Evaluation of the total number of boundary collisions over time at six different car width domains. The maximum number of collisions is 60, as there are 6 evaluations steps (1 before adaptation starts, 5 during) and at each evaluation step 10 evaluations are performed. This cumulative total is averaged over ten initializations for each model. Standard errors are shown with colored bars.

but it also achieves them while being the most safe — it suffers the least number of collisions during adaptation across all these environments.

Comparing RADA ablations, cautious action selection during adaptation proves critical to performance, and `RADA\caution` does much worse than RADA throughout. Domain randomization and pretraining replay have relatively smaller impacts on the performance after ten adaptation episodes, but we will show that they impact training speed and stability.

Training directly on the target domain (`train-on-target`), aside from naturally resulting in more collisions, does not result in the best performance even in terms of rewards. We believe that this is because pretraining on the training car widths has an additional advantage: it sets up a curriculum for training that makes it easier for the agent to solve the exploration problem of learning how to make the right turn.

Finally, `RARL` policies are relatively robust to changes in car width, but perform uniformly worse than `RADA`: they yield lower reward and experience more collisions at all car widths.

**Adaptation speed.** We now plot the results over adaptation time in each target environment for both the average maximum reward and the running total boundary collisions to show adaptation speed in the target environments. Fig 3 shows the average maximum reward over adaptation time for various methods, and at various target car widths. We evaluate at one in-domain car width (0.075) and at five out-of-domain car widths (0.1, 0.125, 0.15, 0.175, and 0.2). Across all six, RADA yields the highest rewards at all times except for car width 0.1. Fig 4 shows similar plots for the average cumulative total boundary collisions over adaptation time. Once again, RADA outperforms all approaches at most domains, with the least overall collisions at all domains except for car width 0.1. Further, as seen in these cases, `RADA\{caution,DR}` adapts more slowly than other approaches, demonstrating the value of multi-domain pretraining. Further `RADA\{caution, replay}` leads to very unstable training, reflected here in the fact that maximum reward over adaptation time does not monotonically increase, unlike the other approaches.

**Epistemic uncertainty.** RADA relies on the fact that a probabilistic dynamics model trained on a population of source domains with varying unknown values of the domain ID $z$, learns to capture

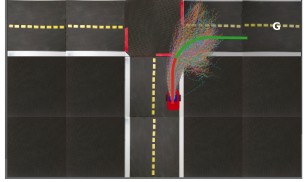 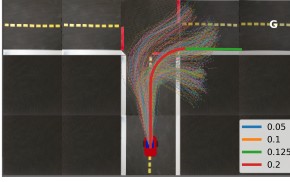

Figure 5: How well does the pretrained dynamics model capture the epistemic uncertainty due to unknown car width? These overhead views present the trajectories predicted by the dynamics model (thin curves in desaturated colors) for a given starting state of the car and a specific action sequence. Overlaid on top of these are ground truth trajectories of the car (thick curves) corresponding to varying car widths (legend in the rightmost panel). See text for details.

the epistemic uncertainty associated with $z$. Fig 5 visualizes how well the dynamics model learns to capture the variation in car behavior due to the unknown car width in our Duckietown setting. For a car in a given state, we evaluate a specific action sequence, and plot the car trajectories predicted by the dynamics model on an overhead view of the environment. These are shown in thin, desaturated curves in the panels in Fig 5. As the panels show, these trajectories originate at the car and spread out as they get farther away, representing the higher uncertainty after more actions have been performed.

On top of these trajectories predicted by the model, we overlay thicker curves corresponding to ground truth trajectories of cars of various widths, executing the same action sequence. In each panel of Fig 5, these ground truth trajectories all lie largely within the support of the predicted trajectory distribution, indicating that the model captures the epistemic uncertainty due to varying car widths.

A particularly illustrative case is in the left-most panel, where the aggressive action trajectory takes the car very close to the corner of the road. This leads to erratic behavior that causes the car to swerve into the side of the road after clipping the corner in some cases, or proceed successfully towards the goal near the center of the road in other cases. The distribution produced by the model's predicted trajectories are similarly bimodal, correctly capturing these two kinds of behavior. Appendix B Fig 7 shows how these predictions evolve over the course of RADA pretraining. Overall, these visualizations demonstrate that the dynamics model successfully learns to/ represent the epistemic uncertainty due to unknown car width.

## 6 DISCUSSION

We have proposed RADA, a new approach to model-based reinforcement learning for safe, quick adaptation of RL agents in new environments with unknown dynamics. RADA relies on two key ideas: transferring knowledge from training in a variety of training environments, and using a maximin notion of risk-aversion during action selection in the target environment. We show in a physically accurate driving environment that RADA performs fast, safe adaptation to learn to drive cars around corners, even when they are up to two times larger than any cars it has driven at pretraining time.

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

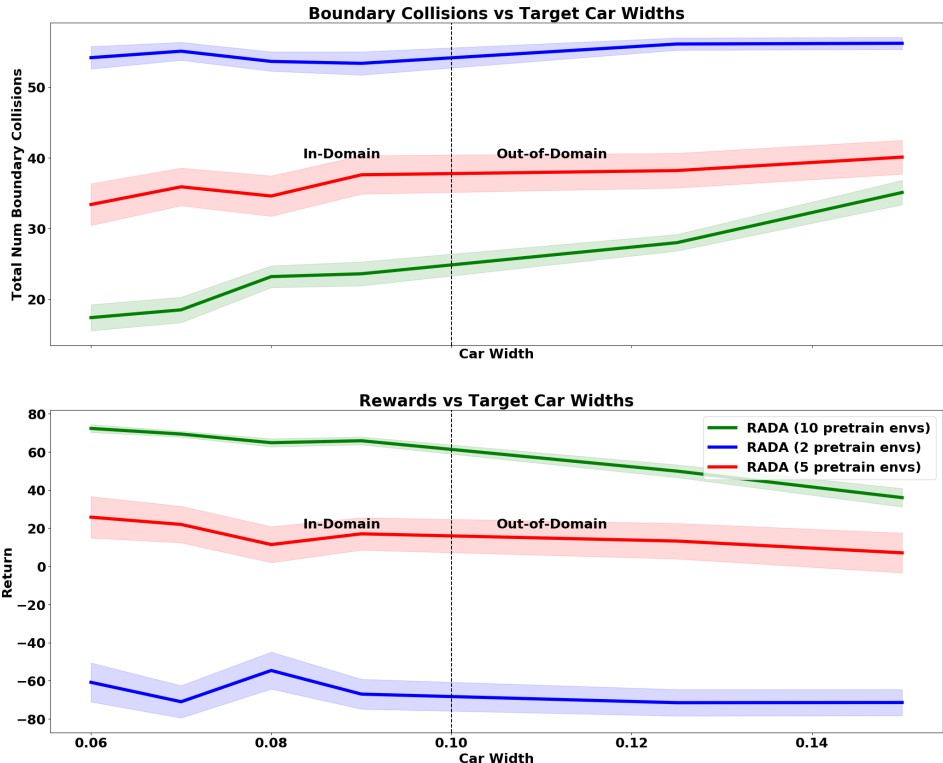

Figure 6: Evaluation of the average maximum reward and average total boundary collisions at different target car widths. (similar to Fig 2, but with a small, fixed number of pretraining environments.)

## A   THE EFFECT OF FINITE PRETRAINING ENVIRONMENTS

In the main paper, we perform experiments where we sample the car width dynamically at the beginning of each pretraining episode, from the full pretraining range (width 0.05 to 0.099), so that there are effectively an infinite number of pretraining environments available to sample from. A more realistic situation might involve having a fixed, small number of pretraining environments, among which we select one at random for each pretraining episode.

We now report the results of preliminary experiments to investigate the feasibility of RADA in this setting. For these experiments, we sample a small number (2, 5, or 10) of car widths from the original pretraining range and limit pretraining to those environments only.

Fig 6 shows these results in the same format as Fig 2. There are significant gains in performance (both reward as well as number of collisions) from 2 to 5 to 10 pretraining environments. At 10 pretraining environments, results approach those reported earlier in Fig 2.

Overall, these results indicate that RADA is feasible when the number of pretraining environments is finite and small.

## B   EVOLUTION OF DYNAMICS MODEL PREDICTIONS DURING TRAINING

In Figure 5, we plotted the trajectory predictions by the fully pretrained dynamics model for a fixed starting state and action sequence. Here, we show how the dynamics model's trajectory predictions improve over pretraining time. To do this, we replot the leftmost panel in Fig 5, but this time using dynamics models from various stages of pretraining. Fig 7 shows the results. At first, the predictions are compeletely random and the predicted trajectories occupy the entire state space. They become more accurate throughout the course of training, approaching the trajectories plotted in the leftmost

Figure 7: How do the dynamics model's predicted trajectories evolve over pretraining time? These plots demonstrate the predicted trajectories for a fixed action sequence at different stages of RADA pretraining, starting at the first episode of training after the initial random rollouts.

image of Fig 5. As the model trains in the pretraining environments, it gradually learns to model the epistemic uncertainty associated with the unknown car width.

