# OpenReview forum: "Hope For The Best But Prepare For The Worst: Cautious Adaptation In RL Agents"
_ICLR.cc/2020/Conference — Reject_

### Official Review · AnonReviewer2 · 2019-10-13
**Official Blind Review #2**

**Rating:** 3

**Review:**

This paper studies the problem of safe adaptation to avoid catastrophic failure in a new environment. It draws intuition from human behavior. The proposed method (risk-averse domain adaptation (RADA)) learns probabilistic model-based RL agents from source domains, and uses them to select actions that has the best worst-case performance in the target domain.

The paper mentions safety-critical applications like auto-driving. However, generally, I don't think black-box models are suitable for these safety-critical applications.

**Experience Assessment:**

I do not know much about this area.

**Review Assessment: Checking Correctness Of Derivations And Theory:**

N/A

**Review Assessment: Checking Correctness Of Experiments:**

I did not assess the experiments.

**Review Assessment: Thoroughness In Paper Reading:**

I made a quick assessment of this paper.

---

### Official Review · AnonReviewer3 · 2019-10-22
**Official Blind Review #3**

**Rating:** 3

**Review:**

This paper tries to address the safe adaptation: given a model trained on a variety of past experiences for some task, train a model learning to perform that task in a new situation while avoiding catastrophic failure.


Pros:
- The idea of training on data from varying quartiles, with the goal of preventing overly-conservative models, is quite intriguing and inspiring.

Cons & Question:
- Motivation:
Cautious exploration or optimizing the best worst-case performance is conflicting with the philosophy of exploration, such as UCB. As stated in the introduction, “enables fast yet safe adaptation within only a handful of episodes.” Intuitively, we can not expect to be safe and fast at the same time. It would be better to discuss why cautious exploration can ensure fast and safe adaption, which would be more interesting. Additionally, in Figure 3, some fast adaption methods, such as MAML, should be compared to be more persuasive.

- Method:
 1. In equation (1), sum_N —> sum_i.
 2. This work formulated safe adaption as minimizing the risk of catastrophic failure. What’s the relationship between “the generalized action score” and “risk of catastrophic failure”? The “generalized action score” is the main difference with PETs. However, it is a little bit hard to follow the idea from “risk of catastrophic failure” to “the generalized action scores”.
 3. “Model-based RL agents contain dynamics models that can be trained in the absence of any rewards or supervision.”
 ”Since dynamics models do not need any manually specified reward function during training, the ensemble model can continue to be trained in the same way as during the pretraining phase.” I am confused about these sentences. Without the reward, what’s the purpose of RL?


- Experiments:
1. As stated in experiments, three meta-learning approaches have been deployed as baselines, including GrBal, RL^2 and MOLe. However, the experimental results are missing. Why meta-learning baselines do not work? Are there any explanations?
2. There are many robust RL baselines, such as
 [1] Pinto L, Davidson J, Sukthankar R, et al. Robust adversarial reinforcement learning[C]//Proceedings of the 34th International Conference on Machine Learning-Volume 70. JMLR. org, 2017: 2817-2826.
It would be better to compare with robust reinforcement learning work since there are no other baselines apart from meta-learning methods.


**Experience Assessment:**

I have published one or two papers in this area.

**Review Assessment: Checking Correctness Of Derivations And Theory:**

N/A

**Review Assessment: Checking Correctness Of Experiments:**

I assessed the sensibility of the experiments.

**Review Assessment: Thoroughness In Paper Reading:**

I read the paper thoroughly.

---

> ### Author Response · Authors · 2019-11-15
> **New robust RL baseline, more metalearning experiments, and clarifications**
>
> Thank you for your extensive comments.
>
> === “Compare against robust RL.” ===
> Thank you for this suggestion. We have now included a new baseline: Pinto et al, Robust Adversarial Reinforcement Learning, 2017 (“RARL”), per your suggestion. Specifically, we train RARL with an adversarial agent that can perturb the motor torques. We pretrain RARL for about 30x as many episodes as RADA (necessary for the model-free approach RARL employs), and evaluate adaptation to new environments similar to our approach. The results are now included in Fig 2, 3, and 4 in the paper. They show that RARL does indeed induce robustness during policy transfer. However, in our experiments, RARL adapts more slowly, yields worse rewards, and leads to more collisions than RADA. Please see Sec 5 and Figs 2, 3, and 4 for more details.
>
> === “Why meta-learning baselines do not work?” ===
> We have tried RL^2, MOLe, and GrBAL (a model-based variant of MAML). Unfortunately, none of these methods work well in our setting. In particular, training these methods has proven extremely unstable in our environment. Following the reviewer's suggestions, we have run experiments to analyze why metalearning fails --- our hypothesis was that it failed due to the large range of training environments. In our new expeirments, we decreased the range of pretraining car widths (from 0.05-0.099 in the paper to 0.05-0.06) in an attempt to stabilize metalearning. We tried training RL^2 and GrBAL once more with reasonable hyperparameter search around the authors’ code defaults. Despite this, neither metalearning approach was able to successfully train. We have added a note on this in the paper.
>
> === “The paper claims fast and safe adaptation, but isn’t fast and safe impossible?” ===
> There is indeed a tradeoff between how safe an agent is, and how fast it can hope to adapt. However, while standard RL and meta-RL approaches do not consider safety at all and therefore provide no ability to trade off safety for speed and vice versa, RADA provides an intuitive way to do this by setting a caution parameter (gamma in Eq 2). It then aims to provide pareto-efficient solutions that pay attention to both safety and adaptation speed, with gamma controlling where on the pareto-frontier the solution is.
>
> ===“I am confused about the sentence ‘Since dynamics models do not need any manually specified reward function during training, the ensemble model can continue to be trained in the same ways as during the pretraining phase.’ Without reward, what’s the purpose of RL?” ===
> We employ a model-based planning approach to RL, involving two steps: (i) a dynamics model is trained that predicts future states given current states and actions, and (ii) then, an action is selected not through a learned policy, but instead by optimizing for actions that produce the most desirable states as predicted by the learned dynamics model. So, the dynamics model is the only component that is learned, and it is task-agnostic and requires no rewards. This is convenient in our setting, since the dynamics models can be trained even in the unseen test environment, where no rewards are provided. RADA exploits this.
>
> === “What is the relationship between the generalized action score and the risk of catastrophic failure?” ===
> Eq 2 defines the generalized action score. This score includes a caution parameter which controls the degree of pessimism with which an action sequence is evaluated during planning with the learned model. For instance, when caution gamma is 50, the generalized action score of an action sequence is the average score of the bottom half of the particles propagated through the model. This would capture any catastrophic failures resulting from those actions, at the cost of ignoring the most successful trajectories that yielded highest reward. As gamma increases, the failures are weighted more relative to the successes. This means that during planning, actions that have even a minor risk of failure are assigned a low generalized action score, and therefore avoided. The generalized action score thus allows control over the degree to which catastrophic failure is avoided.
>
> “sum_N” -> “sum_i” Thank you, we have fixed this and other minor typos now.

---

### Official Review · AnonReviewer1 · 2019-10-25
**Official Blind Review #1**

**Rating:** 3

**Review:**


This paper proposes to adapt RL agents from some set of training environments (which, in the current instantiation, vary in some simple respect) to a new domain. They build on a framework for model-based RL called PETS.

The approach goes as follows:

2-step process
 * train probabilistic model-based RL agents in a “population of source domains”
 * dropped into new environment use “pessimistic exploration policy”

Then at test time, in order to compute estimates for the rewards for each action the authors use a “particle propagation” technique for unrolling through their dynamics model .

The action is chosen by looking at the sum of the 0 through kth percentile rewards.
This is a weird choice. Why are they looking at a sum over quantiles vs a quantile itself?

The claim is that the models from the first stage capture the epistemic uncertainty due to not knowing z.
However, the authors give a too scant a treatment of what these uncertainty estimates really mean.
For example, they appear to only be valid with respect to an assumed distribution over z.
The paper’s experiments however focus in large part on what happens when the model is evaluated
on values of z that were outside the support of the distribution over training domains.
In this case, any benefit appears to be ill explained by the underlying motivation.


The next step here is to finetune the model as data is collected on the new domain.

Authors propose heuristics for this finetuning that include
1. Drawing experiences from the past experiences (under different domains) and
2. “keeping the model close to the original model”, via some sort of regularization presumably.

>>> 	why isn’t the exact nature of how they “keep the model near the original model explained in the text?
	perhaps the authors mean that 1. and 2. are one and the same (1 as  means to achieve 2)
	if this is the case, then the exposition should be improved to make this more clear.


Some important details appear to be missing. For example, how many distinct source domains are seen during pretraining? Do they set z different z for every single episode of pretraining? Some language here is unclear, for example what precisely does an “iteration” mean in the context of the experiments?

The choice to report “average maximum reward” seems strange if what the authors care about is avoiding risk. Can they explain/justify this choice or if not, present a much more comprehensive set of experimental results?

The figures tracking catastrophic failures vs performance resembles those in
“Combating Reinforcement Learning's Sisyphean Curse with Intrinsic Fear”  https://arxiv.org/abs/1611.01211
This raises some question about why they don’t if concerned with “catastrophic events” model them more explicitly.
Else, if the return accurately captures all desiderata, why to we need to count the failures?

In short this is a simpzle empirical paper that makes use of heuristic uncertainty estimates,
including in settings when the estimates have no validity. The writing is reasonably clear
and the ideas are straightforward (which is perfectly fine!). A few of the decisions are unnatural,
a few are ad hoc, and a few details are missing. Overall my sense is that this paper
has some good qualitities, including the clarity of much of the exposition,
but it’s still below the mark to be an impactful ICLR paper.

==========UPDATE=================
I read the rebuttal and am glad that the authors took time to read my review and engage with the criticism as well as try to make some small improvements to the paper, especially exploring the impact on the number of training environments on the results (in the original paper the number of environments available at train time was unlimited). The answers to some of the other questions were less convincing. E.g. the seemingly incoherent objective of summing over the quantiles falls flat. Why should we care more about being a "strict generalization" of some previous algorithm built upon than of having a coherent objective? Overall, I don't think the paper makes it over the bar to accept but I hope the authors continue to improve upon the work and get it into shape where it could be accepted at another strong conference.

**Experience Assessment:**

I have published in this field for several years.

**Review Assessment: Checking Correctness Of Derivations And Theory:**

I carefully checked the derivations and theory.

**Review Assessment: Checking Correctness Of Experiments:**

I carefully checked the experiments.

**Review Assessment: Thoroughness In Paper Reading:**

I read the paper thoroughly.

---

> ### Author Response · Authors · 2019-11-15
> **Epistemic uncertainty analysis, sampling z from a finite set for each pretraining episode, and additional clarifications**
>
> Thank you for this thoughtful review.
>
> === “What do the uncertainty estimates really mean? Do they really capture epistemic uncertainty due to not knowing z?” ===
> Thank you. We have now added visualizations in Sec 5 (particularly Fig 5)  that show the predicted trajectories from our model for a fixed action sequence, and show how it captures the various possible behaviors among car widths encountered in the training data. This provides an empirical validation that the model is indeed able to capture the uncertainty due to unknown car width z. We also include an Appendix B (Fig 7) showing how the model predictions improve during pretraining time until it converges to approximately correctly model the epistemic uncertainty.
>
> === “z different for each episode at training?” ===
> Yes, for the results reported in the paper, we did sample z uniformly at random over the training distribution at the beginning of each episode. We have now added additional results in an appendix showing how RADA performance evolves as a function of the number of available pretraining environments. In particular, we sample a fixed number (2/5/10) of car widths before pretraining and sample uniformly from those during pretraining. Our results indicate that there are significant gains in performance (both reward as well as collision safety) from 2 to 5 to 10. At 10 fixed car widths, results are similar to those originally reported in the paper.
>
> === The review points out that while we propose RADA for safe adaptation to new domains, it still builds on probabilistic models that were learned on training domains, which might perform poorly in unseen domains. ===
> RADA incorporates an inductive bias for “caution”: when dropped into a new environment, a RADA agent starts acting as though the environment is at least as difficult as the most difficult environments it has been trained on. Specifically, it makes the reasonable assumption that actions that rarely caused bad outcomes in training environments are also unlikely to cause bad outcomes in the unseen environments, and selects them. The intuition is that while the new environments are indeed outside the support of what our models could have learned from training environments, they are still within the support of this cautious inductive bias which is built into RADA. Our empirical results establish that RADA does generalize safely to held-out environments.
>
> This built-in bias does not come for free: in environments that are easier than a RADA agent’s training environments (such as smaller car widths in our setting), it would be overly cautious during adaptation to a new environment and thus take longer to reach an optimal policy.
>
> ===“Why sum of 0th through kth quantile, rather than just the k^th quantile”? ===
> Good question, we made this choice so that the RADA would be a strict generalization of the PETS objective: at gamma=0, the RADA objective in Eq 2 exactly matches the PETS objective of Eq 1.
>
> === “Why average maximum reward? If the return accurately captures all desiderata, why do we count the number of collisions (failures)?” ===
> We care about two things: (i) quickly reaching good performance in the target environment, and (ii) safety during the adaptation process. To capture these two desiderata, we report both the average max reward (consistent with Chua et al 2018, which we build on), and the cumulative number of collisions during adaptation. In our experimental setting, collisions correspond to catastrophic states that we can't recover from, and which we aim for RADA to learn to avoid. While return and collisions are closely related in our environment, they do not capture exactly the same thing. In particular, collisions lead to low reward, but low reward does not always mean that a collision occurred. Instead, it might also be due to the agent steering left rather than right, or going about in circles, for example. So it is worthwhile to measure collisions separately to evaluate how safe adaptation is.
>
> === “Keep the model close to the original model. How?” ===
> Yes, we meant that RADA does this by using past experience in training environments during the finetuning stage in the target environment. We have improved the exposition now to clarify this.

---

### Author Response · Authors · 2019-11-15
**New robust RL baseline, epistemic uncertainty analysis, effect of finite pretraining environments, and small fixes**

Thank you for these reviews. In responding to them, we have been able to significantly improve our submission. Specifically, we have made the following changes to the draft:
(i) New Robust RL baseline, as suggested by R3, Robust Adversarial Reinforcement Learning (RARL). This is now included throughout experiments, so that Fig 2, 3, and 4 are all updated. RADA consistently outperforms RARL on all metrics, safety, performance, and learning speed.
(ii) At the end of Sec 5, we have now added a section analyzing and showcasing the ability of RADA’s pretrained dynamics models to represent epistemic uncertainty due to not knowing z, in response to R1's question. Fig 5 provides visualizations.
(iii) We have added Appendix B that further analyzes how the dynamics model predictions evolve over pretraining time until it correctly models the epistemic uncertainty as described above in (ii).
(iv) We have added an appendix A describing new experiments showing the effect of using only a finite number of pretraining environments, rather than sampling from an infinite set at the beginning of each episode, in response to R1. RADA still works well with only about 10 pretraining environments.
(v) We have fixed minor typos and portions of text that reviewers pointed out as being unclear.

We have posted responses to individual reviewers about the points they raised.

---

### Decision · Program_Chairs · 2019-12-19

**Decision:**

Reject

**Comment:**

The work this paper presents is interesting, but it is not quite ready yet for publication at ICLR. Specifically, the motivation of particular choices could be better, such as summing over quantiles, as indicated by Reviewer 1. The inherent trade-off between safety and speed of adaptation and how this relates to the proposed method could also use a clearer exposition.